# Mainly Visual Aspects of Emotional Laterality in Cognitively Developed and Highly Social Mammals—A Systematic Review

**DOI:** 10.3390/brainsci14010052

**Published:** 2024-01-05

**Authors:** Guido Gainotti

**Affiliations:** 1Institute of Neurology, Università Cattolica del Sacro Cuore, 00168 Rome, Italy; guido.gainotti@unicatt.it; Tel.: +39-06-30156435; 2Fondazione Policlinico A. Gemelli, IRCCS (Istituto di Ricovero e Cura a Carattere Scientifico), 00168 Rome, Italy

**Keywords:** emotional laterality, visual emotions, highly developed mammals, methodological problems, ‘right hemisphere hypothesis’

## Abstract

Several studies have shown that emotions are asymmetrically represented in the human brain and have proposed three main models (the ‘right hemisphere hypothesis’, the ‘approach-withdrawal hypothesis’ and the ‘valence hypothesis’) that give different accounts of this emotional laterality. Furthermore, in recent years, many investigations have suggested that a similar emotional laterality may also exist in different animal taxa. However, results of a previous systematic review of emotional laterality in non-human primates have shown that some of these studies might be criticized from the methodological point of view and support only in part the hypothesis of a continuum in emotional laterality across vertebrates. The aim of the present review therefore consisted in trying to expand this survey to other cognitively developed and highly social mammals, focusing attention on mainly visual aspects of emotional laterality, in studies conducted on the animal categories of horses, elephants, dolphins and whales. The 35 studies included in the review took into account three aspects of mainly visual emotional laterality, namely: (a) visual asymmetries for positive/familiar vs. negative/novel stimuli; (b) lateral position preference in mother–offspring or other affiliative interactions; (c) lateral position preference in antagonistic interactions. In agreement with data obtained from human studies that have evaluated comprehension or expression of emotions at the facial or vocal level, these results suggest that a general but graded right-hemisphere prevalence in the processing of emotions can be found at the visual level in cognitively developed non-primate social mammals. Some methodological problems and some implications of these results for human psychopathology are briefly discussed.

## 1. Introduction

An extensive literature has shown that a relation exists between emotions and brain laterality and has proposed three main models (the ‘right hemisphere hypothesis’, the ‘valence hypothesis’ and the ‘approach-withdrawal hypothesis’) that give different accounts of this emotional lateralization. According to the ‘right hemisphere hypothesis’ (e.g., [1]), emotions can be considered a complex adaptive system (complementary to the more phylogenetically advanced cognitive system) and are mainly underpinned by right-hemisphere structures. In contrast, the ‘valence hypothesis’ assumes that a different lateralization may exist for negative and positive emotions, because the former could be mainly represented in the right and the latter in the left hemisphere (e.g., [2]). Finally, the ‘approach-withdrawal hypothesis’ is similar to the valence hypothesis, but maintains that hemispheric asymmetries are not related to the positive or negative valence of the emotional stimuli but to the motivational (approach vs. avoidance) systems that are engaged by these stimuli (e.g., [3,4]). More recently, Ross [5] has proposed a new model of emotional lateralization in humans, called the ‘emotion-type hypothesis’, which assumes that primary emotions and related display behaviors may be modulated by the right hemi-sphere whereas language-dependent social emotions and related display behaviors might be modulated by the left hemisphere.

In recent years, an increasing number of investigations (e.g., [6,7]) have also suggested that the lateralization of emotions is not a uniquely human property but has a long evolutionary history, and that the present state of affairs in humans is the final product of this coevolution.

The complex relationship between human and animal studies on emotional laterality has been critically discussed by Gainotti, who in a first paper [8] focused attention on the methodological problems raised by these investigations and in a second paper [9] tried to circumvent these methodological difficulties with a systematic review of emotional laterality in non-human primates. Results of this systematic review, however, supported only in part the hypothesis of a continuum in emotional laterality across vertebrates.

Data consistent with this assumption was, in fact, obtained only by studies investigating asymmetries in emotional expression at the facial level and in the perception of emotional facial expressions, whereas less convincing results were acquired when reviewing studies concerning gaze asymmetries in the exploration of emotional cues, and disappointing data were obtained in investigations that evaluated possible neurophysiological markers of emotional laterality.

The first group of investigations, indeed, clearly supported the existence of an asymmetry in facial emotional expression (stronger in apes than in monkeys), with greater emotional involvement of the left half of the face, irrespectively of the valence of the expressed emotion and of the method used to evaluate this asymmetry.

In the second group of studies, the most interesting results concerned the presence of gaze asymmetries in looking at personally relevant faces and the influence of these gaze asymmetries on other aspects of position preferences, such as the left-sided cradling bias. This expression denotes the tendency of mothers to hold their baby against the left side of the chest (e.g., [10]), and, according to Salk [11], who first described this phenomenon, could have strong evolutionary origins in other primate species. This statement has been confirmed by Manning and Chamberlain [12] and Manning et al. [13], who have documented this phenomenon in great apes and have assumed that the cradling bias may be linked to the right-hemispheric dominance for emotions, because it exposes the baby’s face to the left visual field of the mother, facilitating a mutual advantage in their emotional interactions. Less convincing results have, in contrast, been obtained by studies that investigated other aspects of the left visual bias, such as its influence on approach and agonistic social interactions.

Finally, a review of putative physiological marker of hemispheric activation during emotional events gave inconsistent and contradictory results, being criticizable from both the empirical and the pathophysiological point of view.

These inconsistent and conflicting results were unsatisfactory, because debates concerning models of emotional laterality and the relationships between human and animal asymmetries are not purely theoretical controversies but can have important implications for the possible relations between emotional laterality and psychopathology (see [14] for a short survey). The ‘right hemisphere hypothesis’ and the ‘valence hypothesis’ allow, in fact, different predictions about the lateralization of various aspects of psychopathology. For instance, some data concerning the neural substrates of major affective psychoses could be consistent with the ‘valence hypothesis’. Some authors (e.g., [15,16,17]) have, indeed, suggested that the major form of post-stroke depression (PSD) might be provoked by left frontal lesions, and others (e.g., [18,19,20,21]) have argued that right-hemisphere lesions are often observed in patients with secondary mania resulting from focal brain injury These results, however, have not been confirmed by more controlled investigations (e.g., [22,23,24,25,26]), and a recent systematic review [27] has even found a significant association between right-hemisphere stroke and incidence of depression in subacute post-stroke periods.

I therefore thought that it could be interesting to expand to other mammals our review of studies investigating different aspects of emotional lateralization, because authors who have claimed that the human laterality of emotions has a long evolutionary history have also assumed that the same models should explain emotional asymmetries across all vertebrates. Furthermore, due to the very large number of heterogeneous investigations dealing with this subject, it was necessary to specify in advance the taxa of mammals and the patterns of emotional behavior that should be included in this review, to draw useful information from this research.

As for the first problem, I thought that the most suitable mammals should be not only cognitively advanced but also very social. Some inconsistencies about emotional laterality obtained in different groups of animals have, indeed, been attributed to the difference between lateralization at the group and at the individual level, and evidence for a role of social behavior in group-level lateralization has been provided by various authors (e.g., [28,29,30]). These reasons led me to think that horses and elephants (within the terrestrial mammals) and dolphins and whales (within the aquatic mammals) could be appropriately included in this new review for general and specific reasons. The general reason was that all these taxonomies of mammals are characterized by large visual fields and by an almost complete decussation of the optical fibers. The specific reason consisted in the fact that the presence in our review of both domesticated and wild animal species could balance the advantages and disadvantages of each of these categories. The main advantage of including wild animals was that this minimized the influence that domestication may have produced on spontaneous animal behavior (e.g., [31,32,33]), whereas the main disadvantage was that it increased the number of observational studies made of wild animals, in comparison with the experimental investigations made of domesticated animals.

As for the second issue, I did not include the facial expression of emotions in the review for two reasons: (a) clear information on this subject has already been gathered by our previous systematic review of emotional laterality in non-human primates [9]; (b) only in primates do we find the anatomical–physiological (muscular and neural) prerequisites that allow satisfactory facial expression/communication of emotions. It may, instead, be more appealing to include in the review other mainly visual aspects of emotional laterality that gave inconsistent results in the previous systematic review of emotional laterality in non-human primates.

## 2. Methods

In this new systematic review, I took into account all studies in the literature that investigated visual aspects of emotional laterality in horses, elephants, dolphins and whales. With this aim in mind, I used PubMed and Web of Science to search for studies that included keywords relevant to visual emotional laterality. The search keywords included terms related to visual aspects of emotional laterality (‘visual emotional laterality’ OR ‘side preferences’ OR ‘visual bias’) AND keywords related to the taxa of social mammals included in the review (‘elephants’ OR ‘horses’ OR ‘dolphins’ OR whales’). During the screening of publications, I also searched within their reference lists to identify additional eligible studies. Since the number of references found with the keyword method was quite low for the “whales” category, in this second step I included in this category also investigations concerning strongly related cetaceans, such as Orca whales

The flow diagram of the review process, reported in Figure 1, is the following:

At the identification stage, 68 records were identified from PubMed and 74 from Web of Science. Twenty-seven additional publications were gleaned from the references of the obtained articles, but 89 duplicate records were removed. The identified reports were, therefore, 80.

At the screening stage, 63 of the identified reports were excluded from further review because they did not concern visually lateralized emotional patterns (*N* = 48), did not consist of empirical studies but of reviews or of theoretical papers (*N* = 10) or concerned reanalyzed samples (*N* = 5). The retained 35 papers were classified in specific categories, to separately take into account different aspects of visual emotional laterality. This classification included three main topics:Visual asymmetries for positive/familiar vs. novel/negative stimuli;Lateral position preference in mother–offspring or other affiliative interactions;Lateral position preference in antagonistic interactions.

The first topic was chosen because, according to many authors (e.g., [34,35]), in many animal species, novel or threatening stimuli are preferentially processed by the left visual field (LVF/right hemisphere) and familiar stimuli by the right visual field (RVF/left hemisphere). However, if right-hemisphere processing of threatening stimuli has often been confirmed, inconsistent or contradictory results have been obtained when the lateralization of positive/familiar stimuli was taken into account (e.g., [36,37,38]).

The second topic was selected because Salk’s [39] suggestion that the left-sided cradling bias observed in humans may have strong evolutionary origins has not only prompted investigations showing that a similar bias also exists in great apes (see Hopkins [40] for review), but also studies aiming to evaluate whether similar left-sided position preferences may be documented in other mammals.

The third topic was chosen because contrasting results can be found on this subject in the animal literature. Some authors (e.g., [41]) have suggested a right-hemisphere dominance when engaging in agonistic responses with conspecifics, contrasting with the left-hemisphere prevalence during positive emotions, but other authors (e.g., [42,43]) have raised important objections to this position.

## 3. Results

### 3.1. General Methodological Aspects and Results of the Studies Included in the Review

Methodological aspects and results of all the studies included in the review have been orderly presented in Table 1, reporting first those concerning the terrestrial categories of ‘horses’ and ‘elephants’ and then those obtained in the marine categories of ‘dolphins’ and ‘whales and other cetaceans’. For each paper included in Table 1, the methodological aspects of the study and the patterns of visual emotional laterality that it could contribute to clarifying have been synthetically described before the main result of the investigation is reported. In the ‘elephants’ category, in which the number of screened records was very low, we decided to also include studies dealing with emotional laterality in which the visual component of emotions was less clear.

This made it possible to analyze data in two steps. The first consisted in separately analyzing methodological aspects and results obtained within each animal category, whereas the second consisted in bringing together results concerning the main topics of the study, irrespectively of the animal category in which they had been obtained. The main advantage of the first step was that it made it possible to evaluate results within the context of the methodology and of the topics more frequently investigated in each animal category, whereas the main advantage of the second step was that it allowed a more general integrated response to the problems raised by each topic.

### 3.2. Results of Studies Conducted within Each Animal Category

#### 3.2.1. Investigations on Domestic or Wild Horses

Most studies belonging to this group (i.e., [44,45,46,47,50,54]), conducted on domesticated horses, evaluated with experimental investigations the visual laterality bias of positive/familiar vs. novel/negative stimuli or people. Results obtained in this group of investigations did not confirm the hypothesis that negative, unfamiliar stimuli are primarily processed by the LVF (right hemisphere) whereas familiar stimuli are processed by the RVF (left hemisphere). Some studies did not find a laterality bias with unfamiliar [44] or negative [54] stimuli. Other studies [45,46,50] found a left-visual-field bias for negative, but no asymmetry for positive, objects, and a final study [47] reported a left visual bias irrespectively of stimulus familiarity.

A second group of studies (i.e., [51,52,53]) conducted systematic observations on the lateralization of mother–foal interactions in domesticated or wild horses. All these studies found that affiliative behavior is significantly left-lateralized and proposed that this “left-eye preference” is due to the right-hemisphere dominance for social processing.

A third small group of observational field studies (i.e., [48,49]) investigated whether lateral biases were present during agonistic interactions in domesticated or feral horses. These studies found that a left-sided bias was present during agonistic interactions both in domesticated and in feral horses.

#### 3.2.2. Investigations on Elephants

Most investigations conducted in this animal category (i.e., [55,56,57,58,61]) were based on systematic observations of the laterality of the unpaired trunk organ during various kinds of exploratory behavior in wild elephants. Since it could be assumed that lateral trunk movements could be influenced by the positive or negative nature of the stimuli to be explored, these studies were tentatively classified as a sub-category of investigations dealing with this general topic. The only population-level lateralization was found in trunk-to-genitals contacts, where right-sided trunk movements prevailed in males touching females [57]. This right-sided bias was attributed to an ipsilateral lateralization of olfactory perception, indicating a right-hemispheric advantage, rather than to an RVF (left hemisphere) prevalence in the processing of social information.

A single observational study [59] investigated the position preferences of mother and offspring wild Asian elephant pairs. Elephant mothers preferred to keep their young in their LVF during slow travelling, whereas a gender effect was observed within the offspring. Elephant sons did, indeed, preferentially keep their mothers in the RVF, while daughters preferred to keep their mothers in the LVF. Finally, one study [60] undertook an anatomical investigation of laterality in the tusk weights of African elephants and found that, in the majority of individuals, the left tusk was significantly heavier than the right one and that individuals with larger overall tusk pairs had a higher degree of tusk laterality. The authors interpreted these findings as due to left-side preferences in tusk use during agonist interactions.

#### 3.2.3. Investigations on Dolphins

Most studies conducted in this animal category (i.e., [64,65,66,67,69,70]) evaluated with experimental methods visual laterality in the processing of positive/familiar vs. novel/negative stimuli or people. Results obtained in these investigations were at variance with the hypothesis assuming that the LVF (right hemisphere) may be involved in negative/unfamiliar stimulus processing and the RVF in the treatment of positive/familiar stimuli. Only one study [70] found that dolphins were visually left-lateralized for negative stimuli, whereas three investigations [65,66,69] found that the RVF (left hemisphere) was preferentially used to explore negative unfamiliar stimuli. One further study [63] found that dolphins used their LVF to look at both familiar and unfamiliar stimuli and another [66] reported no clear visual laterality preference for familiar or unfamiliar stimuli.

A second group of investigations [62,63,68] reported systematic behavioral asymmetries in pectoral fin use and the laterality of flipper-to-body (F-B) rubbing during social interactions. The left side of the body was, indeed, preferentially used in in the first contacts of calves with adults and male dolphins initiated pectoral fin contact with females more frequently with their left pectoral fin.

#### 3.2.4. Investigations on Whales (and Other Cetaceans)

Almost all studies conducted on whales and other cetaceans [71,72,73,74,77,78,79] consisted of systematic observations of calf–mother position preference (visual laterality) or other forms of position preference during social interactions, whereas two studies [75,76] investigated the interactions between visual laterality and the nature of the stimuli to be explored in wild orcas. A general tendency of the calves to position themselves to the right of their mother or of other accompanied adults was almost always found in these investigations, the only exception being a study by Saloma et al. [79] in which calves’ lateralization with regard to their mothers was absent in humpback whales. Furthermore, a more detailed analysis revealed that calf–mother positional asymmetry resulted from the calf’s, rather than by the mother’s, preference to keep the other within the observational LVF [72,73,77]. Furthermore, this position preference was modulated by the situational context (e.g., [74]) and by the calf’s age and gender. The interactions between all these factors were also stressed in two papers [75,76] aiming to analyze the interactions between visual laterality and the nature of the stimuli to be explored in wild orcas.

### 3.3. Results concerning the Main Topics of the Study Irrespectively of the Animal Category in Which They Were Obtained

#### 3.3.1. Visual Asymmetries for Positive/Familiar vs. Novel/Negative Stimuli

Visual asymmetries in the processing of positive/familiar vs. negative/unfamiliar stimuli were assessed in 20 investigations conducted on horses (studies [44,45,46,47,50,55]), on dolphins (studies [64,65,66,67,69,70]), on whales and other cetaceans (studies [75,76]) and (as a tentative sub-category) on elephants (studies [55,56,57,58,61]). Results of these investigations did not confirm the hypothesis assuming a prevalent involvement of the left VF in the processing of negative/unfamiliar stimuli and of the right VF in the treatment of positive/familiar stimuli. Four investigations (i.e., studies [44,45,50,64]) reported, in fact, a greater role for the left VF in the processing of negative/unfamiliar stimuli, but three other studies [65,66,69] reported a prevalent involvement of the right VF in the elaboration of these stimuli. Furthermore, conflicting results were obtained in investigations dealing with visual laterality during predatory behavior in wild orcas, because Karenina et al. [75] reported a prevalent involvement of the right VF and Chanvallon et al. [76] a greater role of the left VF in this kind of activity. Two further investigations [47,64] observed a left VF prevalence for both positive and negative stimuli, and seven other investigations (i.e., studies [44,46,55,56,57,61,67]) found no laterality effects in exploring negative or positive stimuli.

#### 3.3.2. Lateral Position Preference in Mother–Offspring and Other Social Interactions

Lateral position preference in mother–offspring and other social interactions was evaluated in 14 investigations conducted on horses (studies [51,52,53]), elephants (study [59]), dolphins (studies [62,63]) and whales and other cetaceans (studies [71,72,73,74,77,78,79]). In these papers, the authors studied contexts in which mothers and their dependent offspring usually moved side by side on parallel paths. The majority of these investigations (i.e., studies [52,59,72,73,74,77,78,79]) documented the presence of a position preference in mother–offspring interaction and also showed that the young animals played the most active role in placing themselves to the right of their mothers, keeping contact with them with their left visual field. This lateral position preference was observed across all the studied animal categories, with the only exception being study [79]. This preference could, however, be modulated by contextual factors (e.g., [74]), by age (e.g., [77]) and by the gender (e.g., [59]) of the offspring and was also observed in other kinds of young-adult or of general social interactions (e.g., [53,63]). A similar asymmetry was observed during suckling behavior in sperm whales [71] and humpback whales [78], but it is difficult to say whether this asymmetry is also due to a left-visual-field preference.

#### 3.3.3. Lateral Position Preference in Antagonistic Interactions

Lateral position preference during antagonistic interactions (i.e., a tendency to keep the competitor in the LVF) was specifically investigated in only two studies [48,49] on horses. However, data relevant to the same issue can also be derived from results of the anatomical investigation of laterality in the tusk weights made by Bielert et al. [60] in African elephants. Investigations [48,49], in fact, showed that a left-sided bias was present during agonistic interactions both in domesticated and in feral horses, and these data are consistent with the anatomical observation that in most elephants the left tusk is significantly heavier than the right and that individuals with larger overall tusk pairs have a higher degree of tusk laterality. According to Bielert et al. [60], these findings could be due to a left side preference in tusk use during agonist interactions.

## 4. Discussion

If I try to summarize the results concerning the main topics of the study, irrespectively of the animal category in which they have been obtained, an apparent discordance seems to emerge between the results of investigations concerning the different lateral processing of negative and positive visual stimuli and those of studies dealing with lateral position preferences in antagonistic interactions. The first group of studies has, indeed, not confirmed the hypothesis of different involvements for the left and right visual fields in the processing of negative and positive stimuli, whereas the second (smaller) group of investigations suggests that a left-sided bias may be present during agonistic interactions in different animal categories. However, if the results of the first group of studies are clearly inconsistent with predictions based on the ‘valence hypothesis’, because there was not a double dissociation between the processing of positive and negative emotions, those of the second group of investigations are not necessarily at variance with predictions based on the ‘right hemisphere hypothesis’. They may simply suggest that negative emotions are more strongly lateralized to the right hemisphere than are positive emotions and that, therefore, right-hemisphere prevalence emerges more on highly emotionally loaded tasks, such as antagonistic interactions, than on less emotional tasks, such as the processing of familiar/positive vs. unfamiliar/negative visual stimuli.

The model assuming a general but graded right-hemisphere prevalence in the processing of emotions has several advantages. The first is that it is not at variance with models which rightly argue that emotions are experienced, processed and cognitively appraised by both hemispheres (e.g., [5]). The second is that it is consistent with results obtained in studies, conducted in humans and in non-human primates, that have evaluated the comprehension and expression of emotions at the facial or vocal level (see [1,2,80,81,82] for general reviews), because these studies have shown that the right lateralization of emotions is stronger for negative than for positive emotions. The last advantage is that this model is also supported in the present review by results obtained in studies of lateral position preference in mother–offspring interactions, because a position preference of the offspring on the right side of the mother, suggesting a prevalent use of the LVF/right hemisphere for affiliative behavior, has been observed across all the studied animal categories. Furthermore, a similar position preference has also been reported in other kinds of social interactions, making it possible to conclude that results of our review tend to support the ‘right hemisphere hypothesis’ and are at variance with the predictions of the ‘valence hypothesis’.

Two methodological objections to investigations that have studied lateral position preference in mother–offspring or other social interactions, suggest, however, some caution in drawing these conclusions. The first is that most of these studies were based on observational investigations, which are necessarily less controlled than properly experimental studies. The second and more important objection to these investigations is that, when two animals proceed side by side (as in most of these studies), the left VF of one of them corresponds to the right VF of the other. To evaluate the hemisphere involved in this preference, we should, therefore, know which member of the couple plays the most active role in assuming this lateral position. According to Karenina et al., who separately assessed infants’ approaches to their mothers and mothers’ approaches to their offspring in horses [52], elephants [59] and whales [72,73,74], the active role was usually assumed by the offspring, but there were inter-species dissimilarities in different animal taxa. Thus, in horses [52], both mares and foals predominantly kept their pair member in the left visual field during spontaneous approaches, whereas differences in the active roles of mothers and kids were observed in elephants [59] and whales [72,73,74]. In elephants, the active role was assumed by mothers, who preferentially kept their offspring in their left VF (and therefore on their left side) at the population level, whereas in whales individual calves remained significantly longer on their mother’s right side as a result of the calves’ preference to keep the mother in their left VF. Furthermore, these variations were influenced by contextual factors and the age and gender of the offspring in different animal species. Thus, in elephants [59], sons preferentially kept their mothers in the right visual field (in agreement with the mother’s preference to keep the offspring on her left side), while daughters preferred to keep mothers in the left visual field.

Another more general source of caution is that, in most studies included in this review (i.e., in studies [44,46,51,52,53,54,57,63,64,65,66,67,69,72,76]), the expression ‘prevalent use (or involvement) of the right (or left) visual field’ to denote visual asymmetry is more or less systematically replaced by the expression ‘prevalent use (or involvement) of the right (or left) eye’. This expression is often used in many other similar or different animal studies (see [9,83] for a short discussion), but is rather inappropriate, because in all these mammals the partial crossing over of optic nerve fibers at the optic chiasm allows the visual cortex to receive information coming from the same visual field from both eyes. Therefore, even if, according to some authors (e.g., [84,85]), contralateral optic fibers have a larger diameter, more myelin, and transmit sensory information faster than ipsilateral ones, some doubts may emerge about the exact methodology used to identify the presence of visual asymmetry and the amount of hemispheric asymmetry really documented in some of these studies.

On the grounds of these methodological caveats, it seems, therefore, possible to conclude (a) that caution is needed in evaluating the results of this review and (b) that the results are more consistent with the model assuming “graded right-hemisphere dominance” than with the “valence model” assuming different lateralizations of negative and positive emotions.

## Figures and Tables

**Figure 1 brainsci-14-00052-f001:**
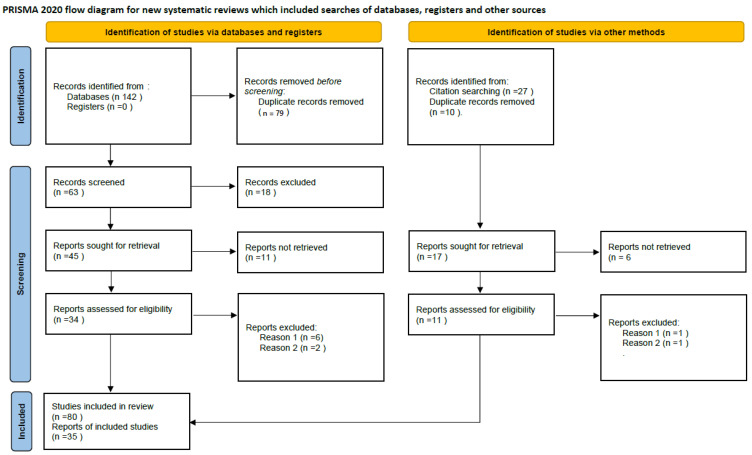
Flowchart of study search and selection.

**Table 1 brainsci-14-00052-t001:** Synthetic descriptions of investigations that studied aspects of emotional laterality in cognitively developed and highly social non-primate mammals.

Investigations on Domestic or Wild Horses		
[44]	Larose et al. (2006)	In this experimental study, the authors evaluated in 65 domesticated horses (*Equus caballus*) whether lateral biases were characteristic of their visual behavior on a novel object test. Horses ”showed no eye preference” to view the stimulus, but the more emotional animals spent more time looking at the novel object with the left eye.
[45]	Austin and Rogers (2007)	In this experimental study, the authors assessed whether 30 domesticated horses (*Equus caballus*) showed greater reactivity to a novel stimulus presented in the left or the right monocular visual field. Horses tested initially on the left side exhibited greater reactivity for left approach, whereas those tested on the right side first displayed no side difference in reactivity.
[46]	De Boyer et al. (2008)	This study evaluated experimentally whether visual laterality in domestic horses differs with objects’ emotional value. They presented to 38 Arab mares three objects associated with positive and negative situations. Horses “used preferentially their left eyes” when looking at the negative object, whereas no asymmetry was found for the positive object.
[47]	Farmer et al. (2010)	This study investigated experimentally visual laterality in domestic horses (*Equus caballus*) interacting with humans. They assessed whether 55 domestic riding horses and ponys entered a chute to the left or right of a person unknown to them. Both groups “showed left eye preference” for viewing the person, regardless of training and test procedure.
[48]	Austin and Rogers (2012)	These authors conducted observational field studies in two groups of 20 and 54 Australian feral horses to determine whether visual lateralization during antagonistic interactions is a characteristic of Equus caballus as a species or results from handling by humans. Horses of the first group had been feral for two to five generations and those of the secong group for 10–20 generations. In both groups, left-side biases were present during agonistic interactions and in reactivity and vigilance.
[49]	Austin and Rogers (2012)	These authors extended their studies on the lateralization of agonistic and vigilance responses to 33 Przewalski horses (*Equus przewalskii*) living under natural social conditions on a large reserve in France. They showed that Przewalski horses exhibit lateralization of agonistic and vigilance responses even more strongly than feral horses, deducing that ancestral horses had similar lateral biases.
[50]	Smith et al. (2016)	In this experimental study, the authors investigated functionally relevant responses to negative (angry) and positive human facial expressions of emotion in 28 domestic horses. Horses showed a left-gaze bias towards negative human facial expressions, but no lateralized response to positive expressions.
[51]	Farmer et al. (2018)	This study systematically evaluated laterality in affiliative interactions in 31 riding horses and mini Shetland ponies (*Equus caballus*). They showed that affiliative behavior (approaching another horse with the left or the right eye) is significantly left-lateralized, suggesting that right-hemisphere specialization in horses is not limited to the processing of stressful or agonistic situations, but rather appears in all social interactions.
[52]	Karenina et al. (2018)	This author conducted observational studies on the lateralization of mother–foal interactions in Przewalski’s horses (*Equus ferus przewalskii*) living in their natural habitat in Mongolia. Lateral position preferences during mare–foal spontaneous reunions were used as a behavioral marker of visual lateralization. Preferences were separately assessed for foals’ approaches to their mothers and mares’ approaches to their foals. Both Przewalski’s foals and mares showed stronger “preference for the left eye use”, predominantly keeping their pair member in the left visual field.
[53]	Inhoue (2019)	This study conducted a drone observation of 23 adult feral horses to evaluate their lateral position preference in grazing. They found that horses form a localized spatial relationship with their nearest neighbor, who is located significantly more frequently to the left than to the right rear of a target individual. The author proposed that this relationship is caused by a “left-eye preference” due to the right-hemisphere dominance for social processing.
[54]	Baragli et al. (2021)	This study conducted an experimental investigation of 77 Italian saddle horses to determine their visual lateralization when inspecting an unfamiliar and unexpected stimulus. They found that horses primarily inspected the balloon with one eye and had a preferred eye to do so, but did not find an eye preference at the population level, concluding that laterality depends on the sample population and testing context.
**Investigations on Elephants**		
[55]	Martin and Niemitz (2003)	This study video-recorded laterality of the unpaired trunk organ in 41 wild Asian elephants (*Elephas maximus*), collecting data on three feeding-related trunk movement categories. While all individuals showed significant bias in grasping, only some animals showed significant bias for retrieval and moving away. Overall, there was no population-level side preference, suggesting lateralized visual control over the task, for any of the trunk movements
[56]	Haakonson and Semple (2008)	This study investigated the laterality of trunk use in eight captive Asian elephants (*Elephas maximus*), quantifying side preference in four different trunk movements: feeding, sand spraying, self-touching and swinging. No overall population-level side bias was seen for any of the four trunk movements.
[57]	Keerthipriya et al. (2015)	These authors examined side preferences in trunk and forefoot movement during feeding in 208 wild Asian elephants. They found no population-level side preference, suggesting lateral visual control over the task.
[58]	Giljov et al. (2017)	These authors studied the lateralization of trunk movements in wild Asian elephants during feeding, trunk-to-mouth contacts and trunk-to-genitals contacts. No side preference at the population level was found in trunk movements during feedind and trunk-to-mouth contacts, but a population-level lateralization was found in trunk-to-genitals contacts, where right-sided trunk movements prevailed in males touching females. The authors attributed this right-sided bias in trunk-to-genitals contacts to ipsilateral lateralization of olfactory perception, indicating a right-hemispheric advantage in the processing of social information.
[59]	Karenina et al. (2018)	These authors investigated the visual preferences of mothers and offspring in 44 wild Asian elephant female young pairs. Elephant mothers preferred, at the population level, to keep the young in their left visual field during slow travelling, whereas a gender effect was observed within the offspring, because sons preferentially kept their mothers in the right visual field, while daughters preferred to keep their mothers in the left visual field. Furthermore, both sons and daughters preferentially kept the familiar older young in the left visual field.
[60]	Bielert, et al. (2018)	This study undertook an anatomical investigation of laterality in the tusk weights of 683 African elephants (*L. africana* and *L. cyclotis*). They found that in the vast majority of individuals the left tusk was significantly heavier than the right and that individuals with larger overall tusk pairs had a higher degree of tusk laterality. The authors interpreted these findings as due to a left side preference for tusk use in rooting, stripping bark and during agonist interactions.
[61]	Lefeuvre et al. (2021)	This study investigated laterality in trunk movements during exploratory behavior under constrained conditions due to the complexity of the task, in six captive African elephants. Both in free exploration and in the experimental condition, elephants showed a general tendency for a right lateralization of most of their behaviors.
**Investigations on Captive or Wild Dolphins**		
[62]	Johnson and Moewe, K. (1999)	This study investigated pectoral fin preference during contact in a group of captive Commerson’s dolphins (*Cephalorhynchus commersonii*). Pectoral touch is suggested to increase social relations among conspecifics. The authors found that the male dolphins initiated pectoral fin contact with females more frequently with their left pectoral fin.
[63]	Sakai et al. (2006)	This study investigated the laterality of flipper-to-body (F-B) rubbing to determine whether wild Indo-Pacific bottlenose dolphins (*Tursiops aduncus*) show asymmetry of eye or flipper use during social behavior. They analyzed 382 episodes of video-recorded F-B rubbings performed by 111 identified individuals and found a population-level left-side bias of the rubber in F-B rubbing, caused by a preference for “use of the left eye”. Dolphins used the left eye significantly more frequently than the right eye during the inquisitive behavior, while they showed no significant bias in flipper use during object-carrying behavior
[64]	Thieltges et al. (2011)	These authors studied, in five captive bottlenose dolphins (*Tursiops truncatus*), the visual laterality of these animals when looking at familiar and unfamiliar humans. Dolphins inspected unfamiliar subjects for longer than familiar subjects, showing that they discriminated between familiar and unfamiliar stimuli, but, at the group level, “preferentially used their left eye” to look at both familiar and unfamiliar humans.
[65]	Siniscalchi et al.(2012)	This study investigated visual lateralization in wild striped dolphins (*Stenella coeruleoalba*) in response to stimuli with different degrees of familiarity. After sighting striped dolphins from a research vessel, the authors presented in a random order different stimuli (fishes, balls, toys) from a telescopic bar connected to the prow of the boat, analyzing the preferential use of right/left monocular viewing during inspection of the stimuli. A “preferential use of the right eye” (left hemisphere) during visual inspection of unfamiliar targets was observed.
[66]	Blois-Heulin et al. (2012)	This study also investigated visual laterality expressed by a group of five common bottlenose dolphins (*Tursiops truncatus*) in response to various stimuli, ranging from very familiar objects (known and manipulated previously) to familiar objects (known but never manipulated) to unfamiliar objects (unknown, never seen previously). At the group level, dolphins “used their left eye” to observe very familiar objects and their right eye to observe unfamiliar objects.
[67]	Yeather et al. (2014)	This study investigated the possibility of visual lateralization in 12 belugas (*Delphinapterus leucas*) and 6 Pacific white-sided dolphins (*Lagenorhynchus obliquidens*), presenting them, during free swim periods, with a familiar human, an unfamiliar human, or no human. Session videos were coded for gaze duration, eye presentation at approach, and eye preference while viewing each stimulus. No clear group-level visual preference for familiar or unfamiliar humans was found.
[68]	Winship et al. (2017)	This study investigated behavioral asymmetries in pectoral fin use during social interactions in a captive population of 27 bottlenose dolphins (*Tursiops truncatus*). While the initiating pectoral fin contact of calves was predominantly performed with their right pectoral fin, the relationship changed in the older age classes, culminating in a left pectoral fin bias in both sub-adults and adults.
[69]	Lilley et al. (2020)	This study examined the laterality of eye use in bottlenose dolphins (*Tursiops truncatus*) and rough-toothed dolphins (*Steno bredanensis*) viewing predictable and unpredictable stimuli. Bottlenose dolphins displayed an overall “right-eye preference”, especially while viewing the unpredictable, moving stimulus, whereas rough-toothed dolphins did not display eye preference while viewing stimuli.
[70]	Mercera and Delfour (2020)	This study investigated visual and motor laterality in five bottlenose dolphins (*Tursiops truncatus*) in spontaneous and experimentally induced emotional contexts. During training sessions, stimuli with positive or negative emotional valences were presented either on the dolphins’ left or right sides. Dolphins were visually left-lateralized during training sessions and reacted more when negative stimuli were presented on their left side than right side during the first stimuli presentation.
**Investigations on Whales (and Other Cetaceans)**		
[71]	Gero and Whitehead (2007)	These authors studied the suckling behavior of sperm whale (*Physeter macrocephalus*) calves using observations from 22 different calves. Previous work assumed that suckling was based on above-water observations of repeated short dives underneath the peduncle of an escort, referred to here as peduncle diving. The authors found that peduncle diving in sperm whale calves is laterally asymmetrical with a bias to the left side of the escorting adult.
[72]	Karenina et al. (2010)	This study investigated the visual laterality of calf–mother interactions in wild beluga whales (*Delphinapterus leucas*), videotaping the social interactions of 29 individually identified wild beluga calf–mother pairs. The individual calves swam and remained significantly longer on a mother’s right side, showing a ”preference to observe their mothers with the left eye”. Frame-by-frame analysis revealed that the positional asymmetry was definitely a result of the calves’, not the mother’s, preference for observing the other with one eye.
[73]	Karenina et al. (2013a)	These authors conducted a further observational study of social laterality in 279 adult–infant pairs of wild beluga whales with an analysis of aerial photographs and direct visual observations of the belugas’ breeding aggregation. A general preference of the calves to position themselves to the right of the accompanied adult was again found.
[74]	Karenina et al. (2013b)	This study investigated the lateral biases in an infant’s position near its mother in wild orcas (*Orcinus orca*). This lateral bias was context-dependent. Observations on mother–infant pairs showed a group-level preference for the infant to be on the mother’s right side when the dyad was far from the boat, whereas this bias reversed at close distance. On the other hand, when infants were socializing near mothers or when they followed older calves, the infants preferred the right side.
[75]	Karenina et al. (2016)	This study explored lateralization in the aerial displays of 60 individually identified resident wild orcas in different behavioral contexts. Side preferences were analyzed in lunging during foraging (related to predatory activity) and breaching (unrelated to this activity). Orcas showed a population-level preference to lunge on the right side when foraging, but not when breaching.
[76]	Chanvallon et al. (2017)	This study investigated the interactions between visual laterality, age, gender and the goal of exploratory behavior in wild orcas (*Orcinus orca*). The results showed a significant “preference for the use of the left eye”, but exclusively in adult females. Adult males displayed more sustained attention than adult females, marked by a higher spatial proximity to divers, slower approaches and longer look durations.
[77]	Hill et al. (2017)	This study explored the lateralized swim positions across environments for beluga (*Delphinapterus leucas*) mother–calf pairs. The results indicated that the calves spent more time on the mothers’ right side than the left both for the Cook Inlet, AK beluga population and for a beluga population in managed care.
[78]	Zoidis and Lomac-MacNair (2017)	This study documented suckling behavior and laterality in nursing humpback whale calves from underwater observations. A pattern of laterality was noted in that all suckling events had a right-side bias. The mother was observed resting in a nearly vertical position and the calf approached on the right side of the mother.
[79]	Saloma et al. (2018)	These authors tried to check whether the newborn calves of humpback whales (*Megaptera novaeangliae*) had a preference to position themselves at the side of their mother. In contrast to what has been described in the literature on other cetacean species, calf lateralization with regard to their position around the mother appeared to be absent in humpback whales.

## Data Availability

Not applicable.

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
