# Peer review of "Mainly Visual Aspects of Emotional Laterality in Cognitively Developed and Highly Social Mammals—A Systematic Review"

_brainsci, 2024, doi:10.3390/brainsci14010052_

Round 1

Reviewer 1 Report

Comments and Suggestions for Authors

This manuscript is an extensive review of whether or not emotions are asymmetrically represented in the brains of highly social (non-primate) animals similar to humans. 35 studies were reviewed that that included horses, elephants, dolphins and whales. The studies were based on assessing reactions to various lateralized visual stimuli. The general conclusion is that “…a general but graded right hemisphere prevalence in the processing of emotions can be found at the visual level in cognitively developed non-primate social mammals.” In reviewing the studies, many contradictory results were found related to how the stimuli were presented that also took into account previous training methods in tamed (domesticated) versus untamed animals when indicated. Also, the confound of individual versus group level lateralization was addressed.  Overall, the review is quite complete and the conclusions justified. My main critique is that the author did not include the Emotion-type hypothesis of emotional lateralization in humans in the Introduction since it is able to encompass the Right Hemisphere, Valence (and Approach-Avoidance) hypotheses and if the author’s conclusions regarding the 35 studies might change if they were evaluated based on the Emotion-type hypothesis.

Reviewer 2 Report

Comments and Suggestions for Authors

Comments on the manuscript “Visual Aspects of Emotional Laterality in Cognitively Developed and Highly Social Mammals. A Systematic Review” submitted to the Animals

General comments

I appreciate the opportunity to review this original and exciting manuscript.

The manuscript is a systematic review of laterality in four major taxonomic groups: horses, elephants, whales, and dolphins. These animals were chosen because they are highly social and cognitively complex. The author selected 35 studies out of 80 previously selected. Visual Asymmetries for Positive/Familiar vs Novel/Negative Stimuli, the preference for the lateral position in agonistic interactions, and the positioning of the pup-mother dyad were analyzed. The author concludes that there is evidence of visual lateralization, but due to some contradictions, caution must be taken when interpreting the results. Another conclusion, perhaps the most important, is that these results are more consistent with the model assuming a “graded right hemisphere dominance” than with the “valence model”.

The manuscript is original, scientifically sound, and relevant to psychological, evolutionary, and neurosciences knowledge fields. The manuscript is well-written, easy to read, and well-organized. The manuscript has great potential to be read by many students and researchers, becoming an important source of knowledge about lateralization in cognitively complex social animals, but not primates.

Although it has many merits, I suggest modifications to increase quality, because I noticed some mistakes and a disarray in the structuring of the manuscript.

Suggestions

Title

The title is not so appropriate to the scope of the manuscript (please, read ahead).

Abstract

The abstract is informative, concise, and complete. However, the author refers to “two main models (the 'right hemisphere hypothesis' and the 'valence hypothesis” (line 8), but in the Introduction (line 30), the author refers to “three main models”. There is a contradiction that must be corrected.

Keywords

It is unusual long terms like “visual aspects of emotional laterality” and “cognitively developed social mammals”. Keywords must be concise, comprehensive, and relevant for cataloging and search engine indexing. I suggest modifying it to more concise terms.

Introduction

The introduction exposes the problem. Models about laterality are exposed and serve as a theoretical basis for what the manuscript proposes to analyze. There is a description of studies on laterality, with an emphasis on what has already been published on primates, including studies by the author himself.

The objectives are well linked with the methods.

Methods

In terms of methods, it is not clear what criteria of methodological rigor and statistical rigor were used by the author for the selection of articles included in this systematic review.

Table 1 is the result of selecting articles that are part of the review. Therefore, it is not a method.

Results

A comment appears on the title and results: the title suggests that the study is about visual lateralization, but in the results on elephants, articles were selected that consider the position of the trunk and the size of the dusk. The proboscis has many functions, but it does not seem to indicate visual lateralization. Smell is more relevant. Likewise, dusk size does not appear to be a proxy for visual lateralization. Except for the pup's position relative to its mother, the other findings on lateralization are not unequivocally linked to visualization. The author could consider changing the title and objectives and removing the solely “visual” approach to lateralization.

Cetaceans include dolphins, whales, and “others”. The separation of analysis between dolphins and whales is intriguing because both groups belong to cetaceans. What is the advantage of separating the results between these two taxonomic groups for understanding the text?

I have doubts about whether there is any advantage because later in the text, the analysis is categorized into the three types of behavior or positioning that suggest lateralization. Therefore, the question can be broader: what is the advantage of showing the results by taxonomic groups (items 3.1.1, 3.1.2, 3.1.3, 3.1.4) and then re-presenting them by type of behavior or positioning (items 3.2.1, 3.2.2, 3.2.3, 3.2.4)?

In part 3.2, the findings of the studies that were presented by taxonomic level (3.1) are again presented. Therefore, I think there is an unnecessary redundancy in the article when it first presents the data at the taxonomic level, presenting the evidence (strong or weak) of lateralization. Next, evidence of behavior or positioning indicating laterality is presented, mentioning which studies and taxonomic groups were studied in each finding.

It seems to me that the focus of the results and discussion is the evidence of lateralization observed in types of behavior and positioning in non-primate mammals. The articles with the taxonomic groups serve as a backdrop for the discussion about which model could best explain lateralization in cognitively complex social animals, other than primates. Reinforcing my point of view, there is no specific focus in the discussion on the evolutionary, ecological, and ontogenetic pressures of lateralization in the mentioned species.

Discussion

In the discussion, in line 503, the author tries to summarize the result with the plural subject (We), but the manuscript is by just one author.

Between lines 569 and 588, a discussion is introduced on the implications of the findings of the present study, in an approach aimed at human cognitive and emotional disorders. There is no connection whatsoever between the review findings and psychic disorders, neither in the objectives and other parts developed in the manuscript nor the discussion. Therefore, it is a part of the text that is loose and meaningless.

References

It seems good.

Round 2

Reviewer 1 Report

Comments and Suggestions for Authors

The manuscript is definitely improved. However, the Emotion-type hypothesis is only indirectly referred to in the Discussion whereas it should also be included in the introduction as well.

Author Response

Reply Reviewer 1

The manuscript is definitely improved. However, the Emotion-type hypothesis is only indirectly referred to in the Discussion whereas it should also be included in the introduction as well.

Response to the reviewer

I have also referred in the Introduction on the Emotion-type hypothesis